# Intra-Patient Genomic Variations of Human Papillomavirus Type 31 in Cervical Cancer and Precancer

**DOI:** 10.3390/v15102104

**Published:** 2023-10-17

**Authors:** Gota Kogure, Kohsei Tanaka, Tomoya Matsui, Mamiko Onuki, Koji Matsumoto, Takashi Iwata, Iwao Kukimoto

**Affiliations:** 1Department of Obstetrics and Gynecology, Showa University School of Medicine, Tokyo 142-8666, Japan; gk_obgyn@med.showa-u.ac.jp (G.K.); monuki@med.showa-u.ac.jp (M.O.); matsumok@mui.biglobe.ne.jp (K.M.); 2Department of Obstetrics and Gynecology, Keio University School of Medicine, Tokyo 160-0016, Japan; kohsei.tnk@gmail.com (K.T.); tomoya.nc.us@gmail.com (T.M.); iwatatakashi@1995.jukuin.keio.ac.jp (T.I.); 3Pathogen Genomics Center, National Institute of Infectious Diseases, Tokyo 208-0011, Japan

**Keywords:** human papillomavirus type 31, variant, phylogenetic analysis, APOBEC3

## Abstract

Human papillomavirus type 31 (HPV31) is detected less frequently in cervical cancer than two major causative types, HPV16 and HPV18. Here, we report a comprehensive analysis of HPV31 genome sequences in cervical lesions collected from Japanese women. Of 52 HPV31-positive cervical specimens analyzed by deep sequencing, 43 samples yielded complete genome sequences of around 7900 base pairs and 9 samples yielded partially deleted genome sequences. Phylogenetic analysis showed that HPV31 variant distribution was lineage A in 19 samples (36.5%), lineage B in 28 samples (53.8%), and lineage C in 5 samples (9.6%), indicating that lineage B variants are dominant among HPV31 infections in Japan. Deletions in the viral genome were found in the region from the *E1* to *L1* genes, but all the deleted genomes retained the *E6*/*E7* genes. Among intra-patient nucleotide variations relative to a consensus genome sequence in each sample, C-to-T substitutions were most frequently detected, followed by T-to-C and C-to-A substitutions. High-frequency, intra-patient mutations (>10%) in cervical cancer samples were found in the *E1*, *E2*, and *E7* genes, and all of them were nonsynonymous substitutions. The enrichment of high-frequency nonsynonymous substitutions strongly suggests that these intra-patient mutations are positively selected during the development of cervical cancer/precancer.

## 1. Introduction

Human papillomaviruses (HPVs) are a large family of small DNA viruses consisting of more than 400 genotypes that differ in the *L1* capsid gene sequence by more than 10% [1]. Based on the epithelial tropism for infection, HPVs can be divided into two groups: cutaneous and mucosal types. Around 13 genotypes of the mucosal type are causative agents of anogenital cancers, especially cervical cancer, and oropharyngeal cancer [2]. HPV type 16 (HPV16) is the most dominant type, accounting for about 60% of cervical cancer cases worldwide, followed by HPV18, which is detected in about 10% of the cases [3]. The remaining causative genotypes vary from region to region of the world. In Europe, HPV33 and HPV45 are the third and fourth most common types, respectively, whereas these are HPV45 and HPV35 in Africa, and HPV52 and HPV58 in East Asia including China, South Korea, and Japan [3].

HPV31 belongs to *Alphapapillomavirus* species 9, which includes HPV16, HPV33, HPV35, HPV52, and HPV58 [4], but is detected relatively infrequently in invasive cervical cancer, accounting for about 4% of cervical cancer cases worldwide [3]. The current nine-valent HPV vaccine contains HPV31 as an antigen, and prevents nearly 90% of cervical cancers, although this vaccine is still not available worldwide, especially in low- and middle-income countries, due to a global shortage [5]. HPV31 has also been traditionally used as a model to elucidate the molecular mechanisms of the life cycle of oncogenic HPVs since human cervical cell lines harboring the complete episomal HPV31 genome were established nearly 30 years ago and made available for laboratory research [6,7,8].

Each HPV genotype is not a single genomic entity but contains a large number of genetic variants that differ by less than 10% in the viral complete genome sequence [9]. Phylogenetically, HPV31 variants have been classified into three lineages (A, B, and C) that are further divided into seven sublineages (A1, A2, B1, B2, C1, C2, and C3) according to the presence of distinct phylogenetic clusters [4]. A recent large-scale study on HPV31 genome sequences from women in the United States has revealed substantial genetic and epigenetic variation in the HPV31 genome and identified a new sublineage, C4 [10]. Despite such extensive work, so far only 56 full-length genome sequences of HPV31 have been registered in GenBank, while more than 600 complete genome sequences are available for HPV16.

The HPV genome exhibits another level of variation called intra-patient single nucleotide variation in the viral genome sequence [11]. This has been most extensively studied for the HPV16 genome in cervical cancer and cervical intraepithelial neoplasia (CIN) lesions, indicating that the HPV16 genome is vulnerable to mutagenesis throughout low-grade lesions (CIN1), high-grade lesions (CIN2/3), and invasive cervical cancer [12,13,14,15]. Among the intra-patient variations observed in the HPV genome, C-to-T base substitutions predominate in CIN1, implicating apolipoprotein B mRNA editing catalytic polypeptide-like 3 (APOBEC3) cytosine deaminases in generating these mutations for antiviral innate immunity [13,16]. Further comparative studies of intrahost genomic variability of oncogenic types (HPV16, HPV18, HPV31, HPV33, and HPV45) revealed the characteristics of APOBEC3-induced mutations in the viral genome [17,18]. In contrast, non-APOBEC3-type mutations detected in the HPV16 genome tend to be enriched in cervical cancer, suggesting that such mutations play a significant role in cancer progression [15].

In this study, to compensate for the lack of viral complete genome sequences, we first attempted to determine the full-length sequence of the HPV31 genome from clinical specimens of Japanese women by employing next-generation sequencing techniques. Phylogenetic analysis of the determined sequences revealed lineage/sublineage distribution of HPV31 in Japan. Furthermore, a detailed analysis of intra-patient variation in viral genomes in individual clinical specimens showed a high level of variability of HPV31 genomic sequences during cervical carcinogenesis.

## 2. Materials and Methods

### 2.1. Clinical Specimens

Cervical exfoliated cells were collected using a cytobrush from Japanese women who visited the Department of Gynecology at Keio University Hospital or Showa University Hospital, as outpatients presenting with symptoms or having been referred by primary care physicians for further examination, between 2018 and 2021. Total cellular DNA was extracted from the cells using the MagNA Pure LC Total Nucleic Acid Isolation kit (Roche, Indianapolis, IN, USA) on a MagNA Pure LC 2.0 (Roche), and subjected to HPV genotyping by PGMY PCR followed by reverse line blot hybridization as described previously [19]. The study protocol was approved by the Ethics Committees at each hospital and the National Institute of Infectious Diseases, and written informed consent for study participation was obtained from each patient.

### 2.2. Next-Generation Sequencing and Bioinformatics Analysis

From clinical samples tested by HPV genotyping, HPV31-positive samples (*n* = 86) were randomly selected and subjected to long-range PCR with KOD One™ PCR Master Mix (Toyobo, Osaka, Japan) to cover the whole-genome sequence of HPV31. The sequences of PCR primers were as follows: HPV31-851F (5′-CTG TAA CTA CAA TGG CTG ATC CAG CAG GTA-3′) and HPV31-1005R (5′-AAC CAT ATC CTC CCC AGT ATC ACT ACT GTC-3′). The amplified DNA was subjected to agarose gel electrophoresis and purified using the Wizard gel purification kit (Promega, Madison, WI, USA). The purified DNA was converted to a DNA library using the Nextera XT DNA sample prep kit (Illumina, San Diego, CA, USA), followed by size selection with SPRIselect (Beckman Coulter, Brea, CA, USA). The multiplexed libraries were analyzed on a MiSeq (Illumina) with the MiSeq reagent kit v3 (150 cycles). The complete HPV31 genome sequences were assembled de novo using the VirusTAP pipeline [20]. The accuracy of the assembled viral whole-genome sequences was verified by read mapping with a Burrows–Wheeler Aligner (BWA) v0.7.12 and a subsequent visual inspection using the Integrative Genomics Viewer v2.3.90.

Nucleotide variations relative to the assembled consensus sequence in each sample were identified using BWA and SAMtools v1.3.1 with in-house Perl scripts as described previously [21]. Variation positions were extracted based on a quality score confidence threshold of Phred quality score > 30 (error probability < 0.001) and defined a position as heterogeneous if the relative read frequency of the variation in each sample was >0.5% and as high-frequency mutation if the frequency was >10%. These intra-patient nucleotide variations were sorted according to six patterns of base substitution (i.e., C-to-T, C-to-A, C-to-G, T-to-A, T-to-G, and T-to-C), or to APOBEC3-type substitutions (i.e., C-to-T or C-to-G in TpC dinucleotides). The overall workflow of the analysis is shown in Figure 1.

### 2.3. Phylogenetic Analysis

The concatenated nucleotide sequences of six genes (*E6*, *E7*, *E1*, *E2*, *L2,* and *L1*) from complete viral genome sequences (*n* = 99), those determined in this study (*n* = 43), and those retrieved from GenBank (*n* = 56) including the reference genomes that represent HPV31 sublineages (A1, J04353; A2, HQ537675; B1, HQ537676; B2, HQ537680; C1, HQ537682; C2, HQ537684; C3, HQ537685), were aligned against each other using MAFFT v7.475 with default parameters. Maximum likelihood trees were inferred using RAxML-NG v1.0.2 under the general time-reversible nucleotide model with gamma-distributed rate heterogeneity and invariant sites (GTRGAMMAI), employing 1000 bootstrap values. Phylogenetic trees were visualized in FigTree v1.4.3.

## 3. Results

### 3.1. Determination of HPV31 Whole-Genome Sequences

A total of 52 specimens positive for HPV31 (single infection, *n* = 28; multiple infections with other types, *n* = 24) were successfully amplified by long-range PCR that covers the whole-genome sequence of HPV31. The number of patients according to cervical disease category was as follows: negative for intraepithelial lesion or malignancy (NILM)/CIN1, *n* = 6; CIN2, *n* = 25; CIN3, *n* = 14; invasive cervical cancer (ICC), *n* = 7 (squamous cell carcinoma [SCC], *n* = 6; adenosquamous carcinoma, *n* = 1). The average age (±standard deviation) in each disease category was as follows: NILM/CIN1, 35.2 (±10.4); CIN2, 36.1 (±11.3); CIN3, 42.8 (±14.5); ICC, 55.4 (±17.9).

The PCR products were converted to short-fragmented DNA libraries by enzymatic tagmentation, followed by Illumina deep sequencing to obtain read sequences covering the viral whole genome. De novo assembly of the read sequences yielded complete genome sequences of HPV31 from 43 samples, the length of which ranged from 7878 to 7967 bp. The remaining nine samples generated shorter, incomplete viral genome sequences, ranging from 3632 to 7064 bp, which showed deletions in the region from the *E1* to *L1* genes, whereas all of them retained the *E6*/*E7* sequence (Table 1).

To determine the variant lineage/sublineage of the complete HPV31 genomes, a maximum likelihood phylogenetic tree was constructed using the nucleotide sequence of the *E6*, *E7*, *E1*, *E2*, *L2,* and *L1* genes. As shown in Figure 2, the presence of distinct clusters including the variant reference genomes allowed for the assignment of variant lineage/sublineage for each HPV31 genome. The deleted genomes were also assigned to lineage/sublineage by conducting phylogenetic analysis of the obtained sequences (Appendix A). Overall, HPV31 variant distribution in the Japanese samples was as follows: lineage A (*n* = 19, 36.5%), lineage B (*n* = 28, 53.8%), and lineage C (*n* = 5, 9.6%). At the sublineage level, sublineage B2 was most dominant (*n* = 18, 34.6%), followed by sublineage A1 (*n* = 14, 26.9%) and sublineage B1 (*n* = 10, 19.2%).

The lineage distribution was similar across the disease categories, but lineage A variants were more frequently detected in ICC (4/7, 57.1%) than in NILM/CIN1 (2/6, 33.3%), CIN2 (8/25, 32.0%), and CIN3 (5/14, 35.7%) (Figure 3). Regarding the lineage distribution in HPV31 single infections, lineage A variants were more frequently detected in ICC (3/5, 60.0%) than in NILM/CIN1 (1/3, 33.3%), CIN2 (2/10, 20.0%), and CIN3 (4/10, 40.0%). Deletion in the viral genome was more commonly observed in ICC (3/7, 42.9%) than in CIN2/3 (6/39, 15.4%), and NILM/CIN1 (0/6, 0%), although the difference was not statistically significant (Fisher’s exact test, *p* = 0.11).

### 3.2. Intra-Patient Variations in the HPV31 Genome

Intra-patient single nucleotide variations with frequencies greater than 0.5% relative to a consensus genome sequence in each sample were extracted from the 43 samples that generated the HPV31 complete genome sequences. Overall, a total of 568 nucleotide variations were detected (Figure 4a), and the number of variations per sample ranged from 1 to 68 (average 13.2, standard deviation 13.2) (Figure 4b). The distribution patterns of variation frequencies were similar across the disease histology (Figure 4c), whereas the average number of variations was increased in ICC (Figure 4d). The variation positions were distributed throughout the viral genome across the histology (Figure 4e).

Classification of the intra-patient variations into six patterns of base substitution revealed that C-to-T substitutions were the most prevalent across the histology, followed by T-to-C and C-to-A substitutions (Figure 5a). These variations were distributed throughout the viral genome, and not enriched in a specific genomic region (Figure 5b).

C-to-T substitutions are mostly derived from APOBEC3-mediated deamination of cytosine in the TpC dinucleotide context [22]. In addition, C-to-G substitutions in TpC motifs are also attributed to APOBEC3 deamination followed by error-prone translesion DNA synthesis [23]. Among the C-to-T and C-to-G substitutions detected in the HPV31 genome, more than half were found in TpC motifs (C-to-T in 140/278 (50.3%) and C-to-G in 15/25 (60%)), implicating APOBEC3 in generating these mutations. These APOBEC3-type substitutions were also distributed throughout the viral genome and detected similarly among CIN1/NILM, CIN2, CIN3, and ICC (Figure 6b), although a relatively high number of such substitutions was observed in some samples of NILM (ID1) and CIN2 (ID8, ID10, and ID20) (Figure 6a).

The intra-patient variations were further sorted according to the viral genomic regions. In CIN1/NILM, CIN2, and CIN3, the highest number of variations was detected in the *E1* gene, whereas it was in the *L1* gene in ICC (Figure 7a). On the other hand, the average frequency of intra-patient variations found in the *E1*, *E2*, and *E7* genes was higher in ICC than in CIN2/3, showing the highest average of variation frequencies of 14.7% in *E2*, 7.3% in *E7*, and 5.2% in *E1* in ICC (Figure 7b).

### 3.3. High-Frequency Mutations in the HPV31 Genome

To further explore the biological relevance of intra-patient variation for cancer development, high-frequency mutations (>10%) were extracted from the CIN2/3 and ICC samples. This yielded 24 mutations: 6 in *E1*, 8 in *E2*, 1 in *E6*, 3 in *E7*, 2 in *L1*, and 4 in *L2* (Table 2). Interestingly, all of these mutations were non-synonymous substitutions, resulting in amino acid changes or introduction of a stop codon. The same mutation (S348L in *E2*) was detected in two CIN2 samples (ID7 and ID17), while other mutations differed between samples. Regarding the substitution pattern, C-to-T or C-to-G substitutions in the TpC context (i.e., APOBEC3-type substitutions) were predominant (14/24, 58.3%), and were more enriched compared to the distribution of all variations detected (Figure 8).

## 4. Discussion

This study is the first to report HPV31 genome sequences from cervical lesions in Japanese women and to show the prevalence of viral variants (i.e., lineages/sublineages) in the specimens analyzed. The results showed that lineage B (and in particular sublineage B2) is predominant among HPV31 infections in Japan. A recent study indicated that the distribution of HPV31 lineages/sublineages varies by region of the world; sublineage C1 is more prevalent in Africa, whereas sublineages B2 and A2/C2 are more common in Europe and Asia, respectively [10]. With regard to the lineage/sublineage distribution in Asia, our results are broadly similar to those of the previous study, although sublineage A1 variants were more frequently detected in our current report. Phylogenetic analysis of the HPV31 genome identified a cluster of sublineage A2, which is restricted to the variants in Japan (ID8, ID9, ID13, ID33, ID35, ID36, and ID52). Japan-specific variants were also reported for HPV16 [24], HPV18 [25], and HPV58 [26], suggesting the possibility of long-term adaptive evolution of HPVs to the Japanese population [27].

A large-scale analysis of HPV31 genome sequences across cervical cancer/precancer revealed that sublineages A1 (odds ratio [OR] = 1.71), A2 (OR = 2.48), and B2 (OR = 1.89) were associated with a higher risk of developing precancer/cancer than sublineage C3 [10]. Our results also suggest that lineage A variants are overrepresented in cervical cancer samples compared to CIN samples, but the small number of the ICC cases (*n* = 6) prevents a statistically reliable assessment of variant risk for cancer progression. Further efforts to collect HPV31-positive clinical specimens, especially single-infection samples, are needed to reach solid conclusions.

Among intra-patient nucleotide variations detected in the HPV31 genome, C-to-T substitutions were a predominant type of substitution, as previously demonstrated for HPV16, HPV52, and HPV58 [13]. This implies a primary role of cellular APOBEC3 in introducing these changes into the HPV genome. Consistent with a recent study describing intrahost variations of HPV31 during CIN progression [18], APOBEC3-type mutations were found across all disease categories, suggesting that APOBEC3 continues to attack the HPV genome throughout cervical carcinogenesis. Most of the APOBEC3 mutations in the HPV31 genome were at a lower frequency in the overall viral population, but some of them reached high frequency in cervical cancer/precancer specimens.

Other prominent intra-host variations detected were T-to-C and C-to-A substitutions, as also described in previous studies [13,15,28]. T-to-C mutations can be introduced during translesion DNA synthesis with cellular error-prone DNA polymerases, such as DNA polymerase eta, which is known to preferentially cause A-to-G transition when copying undamaged DNA [29]. On the other hand, a potential source of C-to-A mutations is reactive oxygen species (ROS) generated within the cell, because the most common type of genomic damage caused by ROS is G-to-T transversion [30]. Notably, the E7 protein activates the translesion synthesis pathway in human foreskin keratinocytes [31], while the E6 protein increases cellular ROS levels when expressed in HPV-negative cervical cancer cells [32].

We observed that high-frequency mutations were more often detected in ICC (3/4 samples, 75%) than in CIN2 (7/19 samples, 36.8%) and CIN3 (3/14 samples, 21.4%), although the differences were not statistically significant (Fisher′s exact test, *p* = 0.13). This may be due to the older age of the ICC cases compared to the CIN2/3 cases because the longer the infection persists, the more likely intra-patient variation will be selected and reach high frequencies in infected lesions. Surprisingly, all of the 24 high-frequency mutations found in CIN2/3 and ICC were non-synonymous substitutions, resulting in an amino acid change or a premature stop codon, which is consistent with our previous work on HPV16 intra-patient variation [33]. This strongly suggests that these mutations were positively selected for during cancer development, and were not due to a bottleneck event acting on a small viral population to engender genetic drift.

The mechanisms of selection for high-frequency mutations include immune escape provided by mutated viral proteins and the growth advantage of cells having mutated HPV genomes. Of note, high-frequency mutations were found in the E1 and E2 proteins, both of which are essential for viral genome replication, transcription, and persistence [34]. Since E1/E2 are likely expressed consistently during viral persistent infection, amino acid changes in E1/E2 may provide an array of different recognition epitopes for T and B lymphocytes, thereby leading to escape from cellular and humoral immunity. Indeed, HPV16 E2-specific T- and B-cell responses were detected in patients with head and neck cancer, suggesting that E2 mutations may cause immune evasion [35,36,37]. Furthermore, since the dysfunction of E1/E2 relieves their repression of the viral early promoter that drives expression of the oncogenes *E6*/*E7* [38], cells harboring the mutated E1/E2 may confer a proliferative advantage to cells and allow them to outcompete their neighbors that harbor the original viral genome. Our previous study demonstrated such inactivating mutations of E1/E2 in the HPV16 genome from cervical cancer specimens [33]. In addition, disruption of *E2* by viral genome integration into the host genome is frequently observed in cervical cancer [39], and was also the case in the present samples.

Our results suggest that APOBEC3-type mutations in the HPV31 genome tend to be enriched as high-frequency mutations in cervical cancer and precancer. This contrasts with a previous study of intrahost variability of the HPV16 genome, indicating that APOBEC3-induced mutations in the viral genome are less likely to be observed in precancer/cancer cases than in controls (i.e., women with a benign HPV16 infection) [15]. In that study, no significant difference was observed in the prevalence of high-frequency APOBEC3-type mutations (frequencies > 60%) between the CIN3+ cases and controls. Further studies are needed to explain the different genomic variability between HPV31 and HPV16.

## 5. Conclusions

This study provides a comprehensive landscape of intra-patient nucleotide variations in the HPV31 genome during CIN progression. The observed pattern of base substitution of the intra-patient variations indicates a primary role for APOBEC3 in introducing these variations into the viral genome. Although the biological relevance of APOBEC3-type mutations for cancer development remains elusive, the detection of high-frequency APOBEC3-type mutations in *E1*/*E2*/*E7* in cancer specimens suggests that these mutations are involved in cervical carcinogenesis at least in some cases.

## Figures and Tables

**Figure 1 viruses-15-02104-f001:**
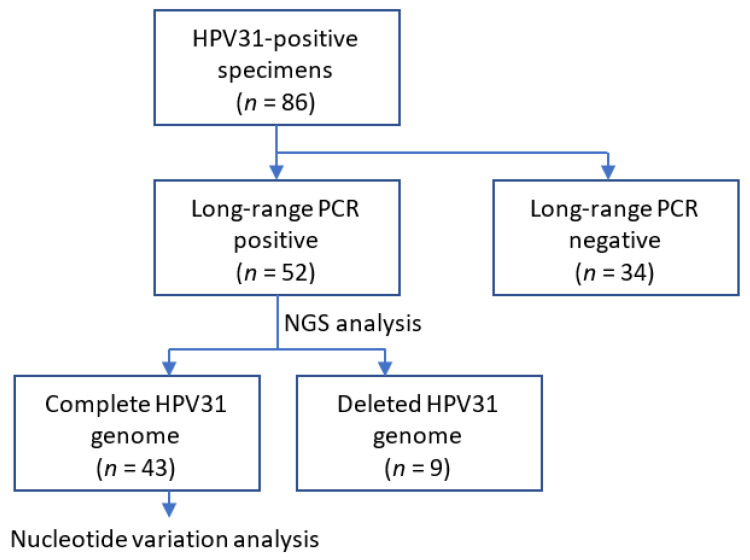
Schematic diagram of the study workflow.

**Figure 2 viruses-15-02104-f002:**
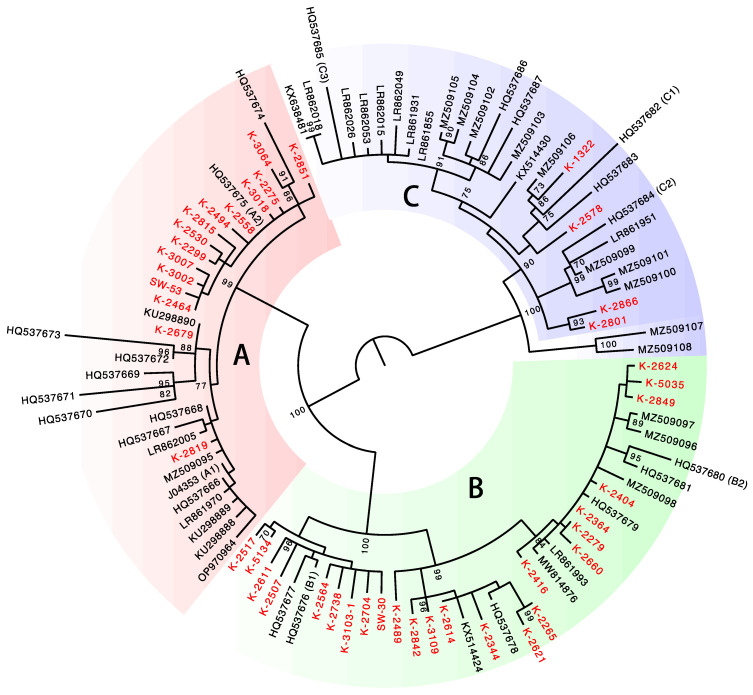
Phylogenetic tree of complete genome sequences of HPV31. Isolates from Japan (*n* = 43) are indicated in red. Bootstrap values > 70% are displayed.

**Figure 3 viruses-15-02104-f003:**
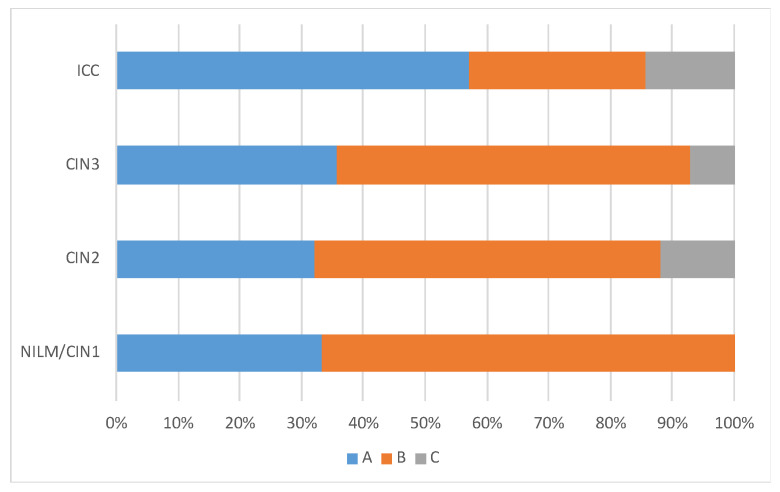
HPV31 lineage distribution according to cervical histology.

**Figure 4 viruses-15-02104-f004:**
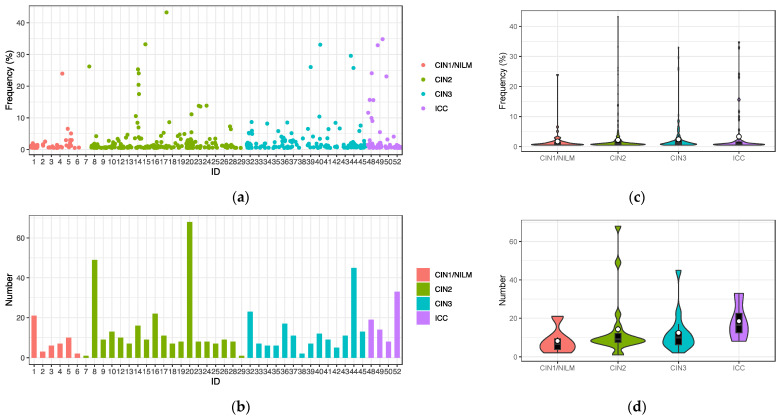
Intra-patient single nucleotide variations in the HPV31 genome. (**a**) Plots of variations with frequencies of >0.5% in individual samples; (**b**) number of variations in individual samples; (**c**) violin plots of variation frequencies according to histology; (**d**) violin plots of numbers of variations according to histology; (**e**) positions of variation in the HPV31 whole genome according to histology. White circles in violin plots represent the average.

**Figure 5 viruses-15-02104-f005:**
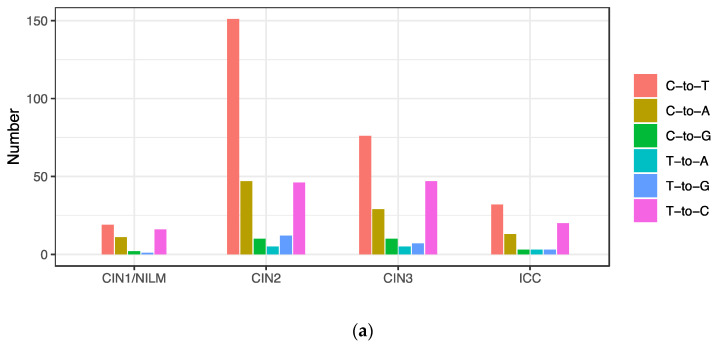
Substitution types of intra-patient variations. (**a**) Number of variations according to base substitution type; (**b**) positions of variation in the HPV31 whole genome according to base substitution type.

**Figure 6 viruses-15-02104-f006:**
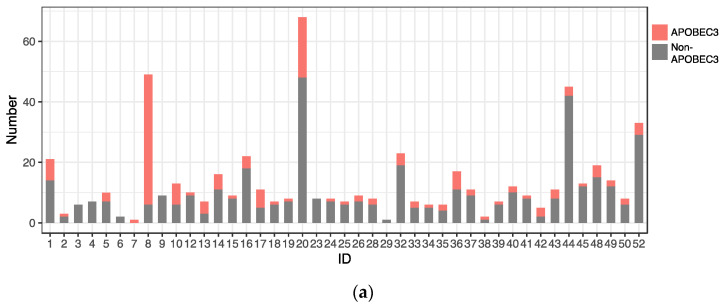
Distribution of APOBEC3-type substitutions. (**a**) Number of APOBEC3-type substitutions in individual samples; (**b**) positions of APOBEC3-type substitutions in the HPV31 whole genome.

**Figure 7 viruses-15-02104-f007:**
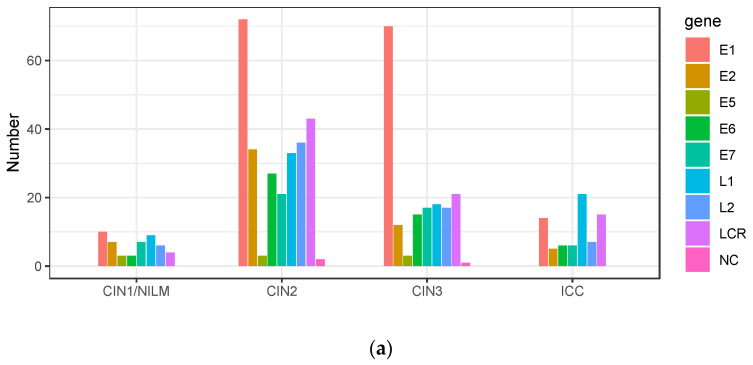
Intra-patient variations according to viral genomic region. (**a**) Number of variations in viral genomic regions according to histology; (**b**) plots of frequencies of variations detected in each genomic region according to disease category. Black bar indicates the average of all variation frequencies in each region. LCR, long control region; NC, non-coding region.

**Figure 8 viruses-15-02104-f008:**
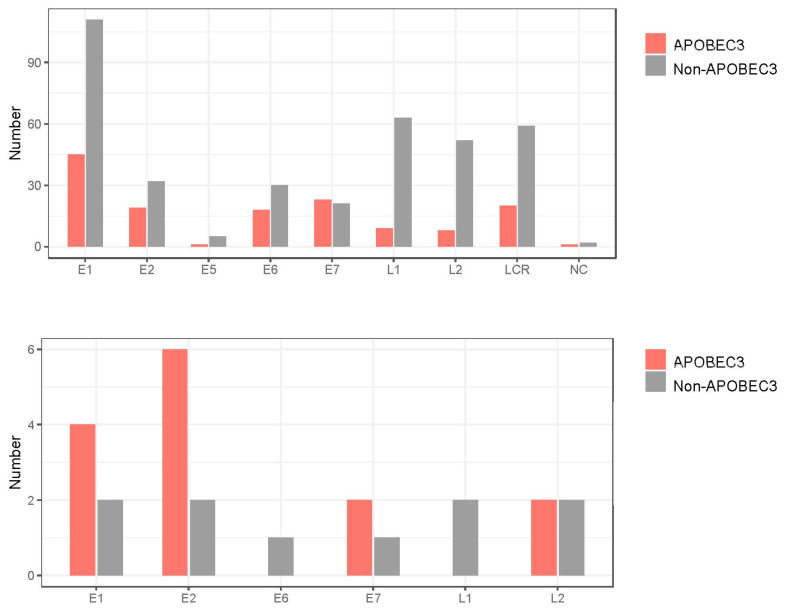
APOBEC3-type substitutions according to HPV31 genomic region. Number of APOBEC3-type substitutions with frequencies > 0.5% (upper panel) and >10% (lower panel). LCR, long control region; NC, non-coding region.

**Table 1 viruses-15-02104-t001:** HPV31-positive specimens analyzed in this study.

ID	Sample	Histology	Age	HPV	Length	Variant	Deletion
1	K-2851	NILM	31	31, 51, 54, 69	7957	A2	-
2	K-2611	NILM	52	31	7892	B1	-
3	K-2404	NILM	34	31, 52	7886	B2	-
4	K-5134	NILM	36	31	7880	B1	-
5	K-2275	CIN1	38	31	7933	A2	-
6	K-2517	CIN1	20	26, 31	7892	B1	-
7	K-2679	CIN2	47	31, 39, 52	7878	A1	-
8	K-3002	CIN2	33	31	7910	A2	-
9	K-3007	CIN2	25	31, 52	7945	A2	-
10	K-3018	CIN2	29	18, 31, 39, 58, 68	7945	A2	-
11	K-2350	CIN2	67	31	4026	A2 ^1^	*E1*/*E2*/*E4*/*E5*/*L2*
12	K-2558	CIN2	47	31, 35, 59	7939	A2	-
13	K-2299	CIN2	43	31, 35, 52	7921	A2	-
14	K-2564	CIN2	60	31	7892	B1	-
15	K-2704	CIN2	26	16, 31, 68	7892	B1	-
16	K-2738	CIN2	25	31	7892	B1	-
17	K-3103	CIN2	38	16, 31, 42	7892	B1	-
18	K-2614	CIN2	52	31	7892	B2	-
19	K-2660	CIN2	38	31	7898	B2	-
20	K-2849	CIN2	24	31, 35, 58	7892	B2	-
21	K-2955	CIN2	26	31	5612	B2 ^1^	*E1*/*E2*/*E4*/*E5*/*L2*
22	K-2546	CIN2	27	31, 52	4493	B2 ^1^	*E1*/*E2*/*E4*/*E5*/*L2*/*L1*
23	K-2416	CIN2	28	16, 31	7892	B2	-
24	K-2364	CIN2	39	31	7880	B2	-
25	K-2344	CIN2	25	31, 42, 52, 68	7892	B2	-
26	K-2265	CIN2	33	31, 58	7889	B2	-
27	K-2642	CIN2	37	31, 51, 56	3821	A ^1^	*E2*/*E4*/*E5*/*L2*/*L1*
28	K-2801	CIN2	34	6, 31, 56, 82	7878	C2	-
29	K-2866	CIN2	32	31, 58	7878	C2	-
30	K-2859	CIN2	36	31	5374	C3 ^1^	*E1*/*E2*/*E4*/*E5*/*L2*
31	K-5104	CIN2	32	31	4257	B2 ^1^	*E1*/*E2*/*E4*/*E5*/*L2*
32	K-2819	CIN3	42	31	7902	A1	-
33	K-2530	CIN3	36	31	7904	A2	-
34	K-3064	CIN3	69	31	7936	A2	-
35	K-2815	CIN3	31	16, 18, 31, 84	7933	A2	-
36	K-2464	CIN3	40	31	7927	A2	-
37	K-2507	CIN3	34	31	7893	B1	-
38	SW-30	CIN3	44	31	7880	B1	-
39	K-2489	CIN3	41	31	7886	B2	-
40	K-2621	CIN3	40	31	7889	B2	-
41	K-2624	CIN3	51	31, 58	7892	B2	-
42	K-2842	CIN3	78	31, 58	7892	B2	-
43	K-3109	CIN3	28	16, 31	7892	B2	-
44	K-5035	CIN3	30	31	7892	B2	-
45	K-2578	CIN3	35	31	7878	C1	-
46	K-2865	SCC	41	16, 31	3821	A1 ^1^	*E2*/*E4*/*E5*/*L2*/*L1*
47	K-2450	SCC	79	31	7064	A1	*E1*
48	K-2494	SCC	50	31	7945	A2	-
49	K-2279	SCC	61	18, 31, 58	7892	B2	-
50	K-1322	SCC	76	31	7878	C1	-
51	SW-58	SCC	51	31	3632	B1 ^1^	*E1*/*E2*/*E4*/*E5*/*L2*/*L1*
52	SW-53	adenosquamous carcinoma	30	31	7945	A2	-

^1^ Lineage/sublineage determined by the incomplete sequences obtained. NILM, negative for intraepithelial lesion or malignancy; CIN, cervical intraepithelial neoplasia; SCC, squamous cell carcinoma.

**Table 2 viruses-15-02104-t002:** High-frequency mutation in the HPV31 genome in CIN2/3 and ICC.

ID	Sample	Position	Frequency (%)	Substitution ^1^	Gene	Variation	Histology
17	K-3103	3735	43.3	**C-to-T**	*E2*	S348L	CIN2
15	K-2704	5447	33.2	T-to-G	*L2*	K426T	CIN2
7	K-2679	3735	26.2	**C-to-T**	*E2*	S348L	CIN2
14	K-2564	1508	25.3	C-to-A	*E1*	W216L	CIN2
14	K-2564	3789	24.0	**C-to-T**	*E2*	S366L	CIN2
14	K-2564	4606	20.4	**C-to-G**	*L2*	D146H	CIN2
14	K-2564	187	17.5	C-to-A	*E6*	R27I	CIN2
24	K-2364	3480	13.9	**C-to-T**	*E2*	R263Q	CIN2
23	K-2416	5315	13.8	C-to-T	*L2*	A382V	CIN2
23	K-2416	2867	13.6	C-to-A	*E2*	V59L	CIN2
20	K-2849	4498	11.1	**C-to-T**	*L2*	E110K	CIN2
14	K-2564	6470	10.6	T-to-G	*L1*	I307L	CIN2
40	K-2621	3797	33.1	C-to-T	*E2*	H369Y	CIN3
44	K-5035	1247	29.6	C-to-A	*E1*	P129Q	CIN3
39	K-2489	1240	26.0	**C-to-T**	*E1*	E127K	CIN3
44	K-5035	6162	25.7	C-to-T	*L1*	T204I	CIN3
40	K-2621	743	10.4	**C-to-G**	*E7*	Q62E	CIN3
49	K-2279	3676	34.8	**C-to-G**	*E2*	W328C	ICC
49	K-2279	3056	32.9	**C-to-T**	*E2*	D122N	ICC
48	K-2494	693	24.1	C-to-T	*E7*	A45V	ICC
50	K-1332	1921	23.1	**C-to-T**	*E1*	E354K	ICC
48	K-2494	2498	15.7	**C-to-G**	*E1*	S546C	ICC
48	K-2494	737	15.6	**C-to-T**	*E7*	Q60 *	ICC
48	K-2494	2438	11.6	**C-to-T**	*E1*	S526F	ICC

^1^ Bold letters indicate APOBEC3-type substitution in TpC motifs. Asterisk indicates a stop codon. CIN, cervical intraepithelial neoplasia; ICC, invasive cervical cancer.

## Data Availability

The HPV31 complete genome sequences determined in this study are available from the DNA Data Bank of Japan with accession numbers LC779559 to LC779578, and LC779596 to LC779618.

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
