# Peer review of "Intra-Patient Genomic Variations of Human Papillomavirus Type 31 in Cervical Cancer and Precancer"

_viruses, 2023, doi:10.3390/v15102104_

Round 1
Reviewer 1 Report
Well written manuscript.
It is not clear from your results or discussion whether there is the same mutation or any new mutation occuring as the tumour progresses from CIN1, 2, 3 and finally invasive cancer, to invoke a multistep process.of carcinogenesis.
It is also unclear as to how these mutations help in evading immune detection.
Both aspects need to be clarified, if they are not mere speculation, but based on your results.
Author Response
Reviewer 1
Well written manuscript.
It is not clear from your results or discussion whether there is the same mutation or any new mutation occurring as the tumour progresses from CIN1, 2, 3 and finally invasive cancer, to invoke a multistep process of carcinogenesis.
Response
Thank you for your favorable comments on our manuscript. We have revised the manuscript according to the reviewer’s comments. Regarding recurrence of mutations, the following sentence was added to the results section:
“The same mutation (S348L in E2) was detected in two CIN2 samples (ID7 and ID17), while other mutations differed between samples.” (line 374 to 376)
It is also unclear as to how these mutations help in evading immune detection.
Response
We have modified the discussion section to account for the possibility of immune evasion due to E2 mutations as follows:
“Since E1/E2 are likely expressed consistently during viral persistent infection, amino acid changes in E1/E2 may provide an array of different recognition epitopes for T and B lymphocytes, thereby leading to escape from cellular and humoral immunity. Indeed, HPV16 E2-specific T- and B-cell responses were detected in patients with head and neck cancer, suggesting that E2 mutations may cause immune evasion [35-37].” (line 445 to 449)
Both aspects need to be clarified, if they are not mere speculation, but based on your results.
Reviewer 2 Report
This study conducted an extensive analysis of human papillomavirus type 31 (HPV31) genome sequences within cervical lesions of Japanese women. Through deep sequencing of 52 cervical specimens that tested positive for HPV31, 43 samples yielded complete genome sequences, while 9 displayed partially deleted genomes. Phylogenetic analysis highlighted the prevalence of lineage B variants among HPV31 infections in Japan. Notably, deletions were identified in the viral genome, primarily within the region spanning from the E1 to L1 genes, although all deleted genomes retained the E6/E7 genes. Intra-patient nucleotide variations were common, with C-to-T substitutions being the most frequent. Remarkably, high-frequency intra-patient mutations (>10%) were identified in the E1, E2, and E7 genes, all of which were nonsynonymous substitutions. These findings strongly suggest that these mutations undergo positive selection during the development of cervical cancer or precancer. While this study is well-executed and pioneering in providing next-generation sequencing data for clinical samples infected with HPV 31 in Japan, minor recommendations include the incorporation of a schematic diagram illustrating the study workflow, encompassing the initial number of clinical samples retrieved, successful PCR amplifications, and samples with successful sequencing data. Additionally, it is suggested to discuss the challenges associated with HPV vaccination, particularly in resource-constrained countries, as highlighted in the following references: DOI: 10.3390/vaccines10091398, DOI: 10.1111/ajco.13513, DOI: 10.3390/curroncol29050303, either in the introduction or results section.
Author Response
Reviewer 2
This study conducted an extensive analysis of human papillomavirus type 31 (HPV31) genome sequences within cervical lesions of Japanese women. Through deep sequencing of 52 cervical specimens that tested positive for HPV31, 43 samples yielded complete genome sequences, while 9 displayed partially deleted genomes. Phylogenetic analysis highlighted the prevalence of lineage B variants among HPV31 infections in Japan. Notably, deletions were identified in the viral genome, primarily within the region spanning from the E1 to L1 genes, although all deleted genomes retained the E6/E7 genes. Intra-patient nucleotide variations were common, with C-to-T substitutions being the most frequent. Remarkably, high-frequency intra-patient mutations (>10%) were identified in the E1, E2, and E7 genes, all of which were nonsynonymous substitutions. These findings strongly suggest that these mutations undergo positive selection during the development of cervical cancer or precancer.
While this study is well-executed and pioneering in providing next-generation sequencing data for clinical samples infected with HPV 31 in Japan, minor recommendations include the incorporation of a schematic diagram illustrating the study workflow, encompassing the initial number of clinical samples retrieved, successful PCR amplifications, and samples with successful sequencing data.
Response
Thank you for your positive comments on our manuscript. As suggested, we have added the workflow as new Figure 1.
Additionally, it is suggested to discuss the challenges associated with HPV vaccination, particularly in resource-constrained countries, as highlighted in the following references: DOI: 10.3390/vaccines10091398, DOI: 10.1111/ajco.13513, DOI: 10.3390/curroncol29050303, either in the introduction or results section.
Response
We have included the vaccination issue in the introduction section as follows:
“The current 9-valent HPV vaccine contains HPV31 as an antigen, and prevents nearly 90% of cervical cancers, although this vaccine is still not available worldwide, especially in low- and middle-income countries, due to a global shortage [5].” (line 43 to 46)